Insulin signaling and pharmacology in humans and in corals

Murthy Meghana Hosahalli Shivananda 1
Jasbi Paniz 1
Lowe Whitney 2
Kumar Lokender 2
Olaosebikan Monsurat 3
Roger Liza 1 4
Yang Jinkyu 5
Lewinski Nastassja 6
Daniels Noah 7
Cowen Lenore 3
Klein-Seetharaman Judith 1 2 8 jkleinse@asu.edu
1 School of Molecular Sciences, Arizona State University , Phoenix, AZ , USA
2 Departments of Chemistry & Physics, Colorado School of Mines , Golden, CO , United States
3 Department of Computer Science, Tufts University , Medford, MA , USA
4 School of Ocean Futures, Arizona State University , Tempe, AZ , United States of America
5 Department of Aeronautics & Astronautics, University of Washington , Seattle, WA , USA
6 Department of Chemical and Life Science Engineering, Virginia Commonwealth University , Richmond, VA , USA
7 Department of Computer Science, University of Rhode Island , Kingston, RI , USA
8 College of Health Solutions, Arizona State University , Phoenix, AZ , United States
Deb Sushanta
Electronic publication date: 2024 Jan 31
Publication date: 2024
Volume: 12
Electronic Location ID: e16804
Received 2023 Jul 27; Accepted 2023 Dec 27
Copyright: © 2024 Murthy et al.
Copyright year: 2024
Copyright holder: Murthy et al.
License: This is an open access article distributed under the terms of the Creative Commons Attribution License, which permits unrestricted use, distribution, reproduction and adaptation in any medium and for any purpose provided that it is properly attributed. For attribution, the original author(s), title, publication source (PeerJ) and either DOI or URL of the article must be cited.
License URL: https://creativecommons.org/licenses/by/4.0/

Keywords: Signal transduction, Non-model organisms, Evolution, Metabolism, Structural biology, Systems biology

Funding: NSF Grants HDR DIRSE-IL OAC-1940169, OAC-1939263, OAC-1939699, OAC-1939795, OAC-1939249, and RAPID 2031614 This work was sponsored by NSF grants HDR: DIRSE-IL OAC-1940169, OAC-1939263, OAC-1939699, OAC-1939795, OAC-1939249, and RAPID 2031614. The funders had no role in study design, data collection and analysis, decision to publish, or preparation of the manuscript.

==============================
Once thought to be a unique capability of the Langerhans islets in the pancreas of mammals, insulin (INS) signaling is now recognized as an evolutionarily ancient function going back to prokaryotes. INS is ubiquitously present not only in humans but also in unicellular eukaryotes, fungi, worms, and Drosophila. Remote homologue identification also supports the presence of INS and INS receptor in corals where the availability of glucose is largely dependent on the photosynthetic activity of the symbiotic algae. The cnidarian animal host of corals operates together with a 20,000-sized microbiome, in direct analogy to the human gut microbiome. In humans, aberrant INS signaling is the hallmark of metabolic disease, and is thought to play a major role in aging, and age-related diseases, such as Alzheimer’s disease. We here would like to argue that a broader view of INS beyond its human homeostasis function may help us understand other organisms, and in turn, studying those non-model organisms may enable a novel view of the human INS signaling system. To this end, we here review INS signaling from a new angle, by drawing analogies between humans and corals at the molecular level.

Rationale

The proposed review aims to conduct an in-depth exploration and comparison of INS signaling pathways in humans and corals, two organisms that exhibit significant evolutionary distance. The necessity of such an investigation arises from the pivotal role INS plays in maintaining glucose homeostasis in mammalian systems, and the potential existence of analogous systems in non-mammalian organisms such as corals. In humans, alterations to INS signaling pathways often result in debilitating metabolic diseases, including diabetes. Conversely, in corals, which are foundational organisms within marine ecosystems, a thorough understanding of potential INS signaling pathways is notably absent. This omission is significant, as corals are currently under considerable threat due to anthropogenic activities and climate change, making insights into their resilience mechanisms vitally important. Bridging knowledge gaps between mammalian and non-mammalian INS signaling has the potential to expand our comprehension of evolutionary biology, provide novel insights into metabolic disease pathophysiology, and afford valuable knowledge for the conservation of coral species.

Intended audience

This review aims to provide valuable knowledge to a diverse scientific audience, encompassing researchers and academics engaged in disciplines such as biochemistry, cellular biology, endocrinology, and marine biology. It seeks to furnish scientists specializing in human metabolism and endocrine disorders with novel insights gleaned from non-mammalian systems. Concurrently, it aspires to equip marine biologists and coral ecologists with a deeper understanding of potential INS signaling within coral biology. Moreover, this review will be beneficial for pharmacologists, particularly those seeking to apply their understanding of human INS signaling to burgeoning fields such as “coral pharmacology.” Educators may find this comparative study valuable as a teaching resource, and students will gain exposure to the application of established biological concepts to innovative contexts. Ultimately, by delineating connections between humans and corals, the review aims to foster interdisciplinary dialogues, encourage collaborative research initiatives, and contribute to both human and coral health.

Evolutionary conservation of INS

INS signaling is an evolutionarily ancient function, as evidenced by the fact that INS-like molecules have been identified in prokaryotes, microbial eukaryotes, insects, invertebrates (including Hydrozoa), and plants (Le Roith et al., 1980; LeRoith et al., 1981; Baig & Khaleeq, 2020). Antibodies raised against human INS recognize INS-like material from unicellular eukaryotes such as Tetrahymena pyriformis, a ciliated protozoan, and Neurospora crassa (Muthukumar & Lenard, 1991; Kole, Muthukumar & Lenard, 1991) and Aspergillus fumigatus, both fungi, and even prokaryotes (LeRoith et al., 1981). The fact that both prokaryotes and eukaryotes synthesize INS suggests that it may play a role in co-evolution.

Conservation at the sequence level is mirrored by conservation at the functional level (Abou-Sabe’ & Reilly, 1978; Le Roith et al., 1980). For example, effects of mammalian INS on E. coli have been described (Abou-Sabe’ & Reilly, 1978; LeRoith et al., 1981); similarly, INS shows metabolic effects on N. crassa cells such as enhanced glucose metabolism, enhanced growth, improved viability, and accumulation of intracellular sodium (Muthukumar & Lenard, 1991; Kole, Muthukumar & Lenard, 1991). INS-like preparations from more primitive organisms have effects on rat cells (Schmidt, Siegel & Creutzfeldt, 1985; Aguan et al., 1994; Cheng et al., 2007). These functional effects are likely achieved through a phosphorylation cascade as shown by the enhanced phosphorylation of specific proteins on serine/threonine and tyrosine residues (Kole, Muthukumar & Lenard, 1991).

Recently, an INS-INSR pair has been described in detail for Acanthamoeba castellanii, an early mitochondrial unicellular eukaryotic organism (Baig & Khaleeq, 2020). Not only did they show typical mammalian INS-induced effects on Acanthamoeba cells, but they also investigated the anti-diabetic drug metformin, and conducted homology modeling of the putative Acanthamoeba INS-INSR pair. This study strongly supports the notion of a high degree of conservation of the INS-INSR pair across billions of years of evolution and pioneers the use of a human antidiabetic drug (metformin) in the context of a primitive organism.

Most recently, remote homologues of INS and INSR have been identified in the stony coral, Pocillopora damicornis (Roger et al., 2022), using a new bioinformatics pipeline for identifying functions in non-model organisms (Kumar et al., 2023) based on remote homology detection suitable for comparison of sequences that show large divergence due to evolutionary separation called HHblits (Remmert et al., 2011). The conservation of INS signaling in corals is also seen in another cnidarian, Hydra, which belong to another phylum of cnidarian. Corals have evolved before the split into Deuterostomia such as humans and Protostomia like the model organisms C. elegans and Drosophila melanogaster 700 million years ago (mya). The Anthozoa-Hydrozoa separation, i.e., the coral-Hydra separation, occurred >500 mya (Khalturin et al., 2019). In Hydra, a receptor protein-tyrosine kinase responding to an INS-like molecule was found to be involved in regulating cell division and differentiation (Steele et al., 1996).

These evidences for evolutionary conservation of INS and its function open the door to comparison of model and non-model organisms. Most of our understanding of the function of INS comes from studies in human as a model organism. We will therefore first review what is known about INS in humans and then compare the findings in detail to one non-model organism, Pocillopora damicornis. Because corals are threatened by extinction, we hope that the extension of our understand from the well-studied human INS pharmacology may provide clues to how we could treat the growing problem of coral bleaching and stimulate a new era of “coral pharmacology”.

Introduction to glucose homeostasis and insulin function in humans

The primary source of energy for most cells in the body is glucose and it is also a substrate for many biochemical reactions (Nakrani, Wineland & Anjum, 2023). Blood glucose levels in the body are maintained and balanced by glucose homeostasis (Da Silva Xavier, 2018), as outlined in Fig. 1. Glucose, as a highly polar molecule, cannot diffuse into the lipid membranes of cells and its transport is therefore facilitated by glucose transporters (GLUTs), a family of 12 members (Navale & Paranjape, 2016). Glucose is phosphorylated upon entering the cell and is broken down through glycolysis, followed by oxidative phosphorylation of pyruvate in the TCA cycle, generating ATP (Fukunaga & Hunter, 1997; Watowich et al., 1999). Alternatively, it is polymerized to glycogen for storage of excess glucose (Nakrani, Wineland & Anjum, 2023). To maintain the balance between these opposing processes resulting in regulated blood glucose levels hormones are produced from a group of multicellular endocrine cells called Islets of Langerhans (Da Silva Xavier, 2018). On average, the human pancreas contains 3.2 million islets. Islets consist of four major types of cells: α-cells, β-cells, δ-cells, and pancreatic polypeptide (PP)-cells (Erlandsen et al., 1976; Da Silva Xavier, 2018). β-cells produce and store insulin (INS) which lowers blood glucose levels, while α-cells produce glucagon (GCG) raising blood glucose concentrations, and δ-cells produce somatostatin (SST) which inhibits the secretion of growth hormones and GCG, while PP cells secrete gastrointestinal and intestinal enzymes (Erlandsen et al., 1976), shown in Fig. 1.

Figure 1 The role of insulin signaling in glucose regulation.

(Left) Regulation of blood glucose in humans. The rise in the blood glucose level releases INS from the pancreas into the bloodstream. This INS stimulates the liver to convert blood glucose into glycogen for storage and SST secreted inhibits GCG secretion. When blood glucose level is low, pancreas release GCG, which causes the liver to turn stored glycogen back into glucose and release it into the bloodstream. SST in this case inhibits INS secretion. (Right) Schematic of Islet of Langerhans architecture. Created with BioRender.com.

INS release is triggered when the glucose concentration in blood rises above 90 mg/dL (5 mM) (Steiner et al., 1967; Fu, Gilbert & Liu, 2013). GLUT1 mediates intracellular glucose transport in the β-cells and triggers the immediate release of INS into the blood from β-cells. INS then binds to insulin receptors (INSRs) on target tissues (Steiner et al., 1967; Xu, Paxton & Fujita-Yamaguchi, 1990; Bremser et al., 1999). This is followed by a cascade of phosphorylation reactions initiated by INSR resulting in mitogenic and widespread metabolic effects of INS such as activation of the phosphatidylinositol-3-kinase (P13K) signaling pathway as described in detail below and references (Xu, Paxton & Fujita-Yamaguchi, 1990; Jones et al., 1991; Vainikka et al., 1994; Fukunaga & Hunter, 1997; Watowich et al., 1999; Hennige et al., 2000; Zhang et al., 2006; Kuo et al., 2007). INS signaling stimulates glucose translocation by INS-responsive GLUT4 and uptake of glucose by target tissues (Steiner et al., 1967; Kawanishi et al., 2000). GLUT4 is found in skeletal muscle, heart, brain and adipose tissues (Navale & Paranjape, 2016). Many putative intra-islet messengers have been implicated in regulating INS secretion, including ATP, Zn2+, γ-aminobutyric acid (GABA), and glucagon-like peptide-1 (GLP-1) (Reetz et al., 1991; Franklin & Wollheim, 2004; El et al., 2021). GABA released from β-cells binds and activates GABAA receptors on α- and δ-cells, which in turn mediates glucose-dependent GCG release and increases SST secretion via activation of INSR on α- and δ-cells, respectively (Xu et al., 2006; Braun et al., 2009). Glucose-dependent insulinotropic polypeptide (GIP) and GLP-1 are secreted in response to ingestion of glucose and amino acids in the gut (El et al., 2021). In the pancreas, binding of GIP and GLP-1 to G protein coupled receptors (GPCRs) such as GLP-1R and glucagon receptor (GCGR) activates adenylyl cyclase/adenylate cyclase 8 (ADCY8) increasing intracellular cAMP signals and stimulates release of INS, GCG and SST (Moreau et al., 2006; Cheng et al., 2007; Fridlyand & Philipson, 2016). GCG from the pancreatic α-cells is the primary proglucagon derived peptide (PGDP) and GLP-1 and GLP-2 are related to GCG as they are co-encoded within the same proglucagon gene (Drucker, 2005). GLP-1 has been reported to improve insulin resistance and increase insulin sensitivity in obese and diabetic humans by modulating endoplasmic reticulum stress response via mTOR signaling pathway inhibition and activation of central GLP-1R (Sinclair & Drucker, 2005; Parlevliet et al., 2010; Jiang et al., 2018). GLP-2 is co-secreted with GLP-1 in the gut but very little is known about its action on glucose regulation in humans. So far, it is clear that GLP-2 is not involved in INS release (Schmidt, Siegel & Creutzfeldt, 1985; Sinclair & Drucker, 2005; Amato, Baldassano & Mulè, 2016). However, when GLP-2 is injected into healthy non-obese humans and diabetic patients, it increases glucagon secretion in plasma (Meier et al., 2006; Amato, Baldassano & Mulè, 2016). In humans afflicted by obesity and type 2 diabetes there is an association between GLP-2 and INS resistance with beneficial effects on glucose metabolism (Amato, Baldassano & Mulè, 2016). The main function of GLP-2 are energy uptake regulation and maintenance of intestinal mucosal integrity, function, and morphology (Amato, Baldassano & Mulè, 2016).

In summary, INS controls the glucose levels in the body and stimulates update of glucose which accumulates and is converted to glycogen and fat within muscles, liver, and adipose tissues (Quesada et al., 2008). GCG counterbalances INS action by activating glycogenolysis and gluconeogenesis in the liver. SST inhibits endocrine hormone secretions. The evidence so far further suggests that GLP-1 is involved in INS release, and less in INS signaling, while the role of GLP-2 in INS signaling has not yet been widely investigated.

Understanding these hormone actions and their downstream signaling pathways is crucial because of the clinical relevance for diabetes. Diabetes is a metabolic disorder in which the body produces less INS or has reduced sensitivity to INS. GLP-1R agonists are well-studied for the treatment of diabetes as they regulate blood glucose by increasing INS secretion and inhibit GCG secretion and appetite (Drucker, 2005). In the standard treatment for advanced diabetes, INS is supplied to compensate (Drucker, 2005). However, the INSR itself has also been considered a potential drug target to stimulate INS signaling by using INS ligands that directly bind to and activate the receptor (Kumar, Vizgaudis & Klein-Seetharaman, 2021). Other proteins such as GLUT-4 are also drug targets (Bouché et al., 2004), and one of the most successful drugs to treat diabetes, metformin (Bouché et al., 2004), likely has multiple targets, including the INSR (Bouché et al., 2004).

Survey/search methodology

Our comprehensive literature search was conducted using an array of scientific databases, including but not limited to PubMed, Google Scholar, and Web of Science. This multi-platform approach was implemented to ensure the broadest possible coverage of available literature.

Our search strategy involved the use of key terms and phrases, carefully chosen for their relevance to the subject matter. These included “insulin signaling,” “insulin receptor,” “corals,” “glucose homeostasis in corals,” “coral bleaching,” “coral pharmacology,” and “coral metabolism.” Additionally, we used Boolean operators to refine our searches, combining terms such as “insulin AND corals,” “insulin signaling AND corals,” “insulin receptor in corals,” and “insulin signaling in corals AND diabetes in humans”.

To ensure an unbiased and objective review, we utilized a set of pre-determined inclusion and exclusion criteria. Inclusion criteria encompassed peer-reviewed research articles, reviews, reports, and meta-analyses published in English within the past two decades. A particular emphasis was placed on studies published within the past 5 years to maintain a focus on current and emergent findings. We sought to include a comprehensive range of literature, focusing not only on recent studies but also incorporating seminal works that have significantly contributed to the field, regardless of their publication date. Inclusion criteria encompassed peer-reviewed research articles, reviews, reports, and meta-analyses published in English, with a strong emphasis placed on studies that have had a substantial impact on the field. While a special emphasis was placed on literature published within the past 5 years to highlight the most current and emergent findings, we also included older literature, particularly those fundamental to our understanding of insulin signaling in humans and corals.

Exclusion criteria involved literature that did not directly pertain to insulin signaling in either humans or corals, studies not subjected to peer-review, and non-English publications. Furthermore, we considered the citation count of each study, using it as a metric of its impact within the scientific community. Following the identification of relevant literature, each publication underwent a meticulous review process. The gathered information was then synthesized and critically evaluated to highlight the current understanding of insulin signaling in humans and corals, identify gaps in knowledge, and provide a comprehensive view of potential future research directions.

Systems biology of INS in humans: INS-related signal transduction cascades

The complexity of the INS-related signal transduction pathways is depicted in Fig. 2, and the proteins involved are listed in Table 1. Proteins are separated by pathway involvement based on whether the pathways are initiated by GCG, GLP-1, INS, or SST. Note that GLP-2 is omitted from Fig. 2 due to our gaps in knowledge of how it interfaces with the action of the other hormones and downstream signaling pathways. Full protein names, UniProt entry names, PDAM ID, E-value, P-value, sequence identity, similarity and references regarding biological function are provided and all protein isoforms of a related gene are listed in Tables 1 and 2 are discussed together. A total of 75 proteins (excluding isoform counting) are involved across the three pathways in humans (GCG and GLP-1 receptor binding events are treated as one pathway for simplicity). Of these, 17 proteins function exclusively within the GCG/GLP-1 signaling pathway, while 36 are exclusive to canonical INS signaling, and 10 are exclusive to the SST pathway. Eleven proteins are involved in two of the pathways, while only one protein is ubiquitous to all three pathways AKT (RAC-α/β/γ serine/threonine-protein kinase). Inhibitory signaling is found in GCG and INS pathways, while only stimulatory signaling is maintained in the SST pathway.

Figure 2 Overview of insulin related signaling pathways.

Conservation of INS-related signaling pathways. Proteins (and their associated isoforms as detailed in Table 1) are represented by their gene name. Proteins exclusive to the GCG signaling pathway are colored in blue and proteins exclusive to INS signaling are colored in orange, while those of the SST signaling pathway are colored pink; proteins involved in two or more pathways are shaded light green. Created with BioRender.com.

Table 1 Proteins related to human insulin signaling.

Full list of proteins involved in INS and INS-related signaling pathways (i.e., GCG and SST signaling), detailing associated pathway, full name, UniProt entry name, and references to biological function.

Gene	Protein	UniProtKB	Reference	
Glucagon	
ADCY8	Adenylate cyclase type 8	P40145 (ADCY8_HUMAN)	Leech, Castonguay & Habener (1999)	
CALM*	Calmodulin-1
Calmodulin-2
Calmodulin-3	P0DP23 (CALM1_HUMAN)
P0DP24 (CALM2_HUMAN)
P0DP25 (CALM3_HUMAN)	Tsang et al. (2006) and Chattopadhyaya et al. (1992)	
CBP	CREB-binding protein	Q92793 (CBP_HUMAN)	Zhang & Bieker (1998)	
CREB1	Cyclic AMP-responsive element-binding protein 1	P16220 (CREB1_HUMAN)	O’Donovan et al. (1999)	
CRTC2	CREB-regulated transcription coactivator 2	Q53ET0 (CRTC2_HUMAN)	Iourgenko et al. (2003)	
GCG*	Pro-glucagon	P01275 (GLUC_HUMAN)	Orskov, Wettergren & Holst (1993)	
GCGR*	Glucagon receptor	P47871 (GLR_HUMAN)	MacNeil et al. (1994)	
GLP-1*	Glucagon-like peptide 1	P01275 (GLUC_HUMAN)	Orskov, Wettergren & Holst (1993)	
GLP-1R*	Glucagon-like peptide 1 receptor	P43220 (GLP1R_HUMAN)	Thorens et al. (1993)	
GNAQ	Guanine nucleotide-binding protein G(q) subunit alpha	P50148 (GNAQ_HUMAN)	Alvarez-Curto et al. (2016)	
GNAS	Guanine nucleotide-binding protein G(s) subunit alpha isoforms short
Guanine nucleotide-binding protein G(s) subunit alpha isoforms Xlas
Neuroendocrine secretory protein 55	P63092 (GNAS2_HUMAN)

Q5JWF2 (GNAS1_HUMAN)

O95467 (GNAS3_HUMAN)	Pak, Pham & Rotin (2002);
Montrose-Rafizadeh et al. (1999) and Zill et al. (2002)	
IP3R	Inositol 1,4,5-trisphosphate receptor type 3	Q14573 (ITPR3_HUMAN)	Holz et al. (1999)	
PFKFB1	6-phosphofructo-2-kinase/fructose-2,6-bisphosphatase 1	P16118 (F261_HUMAN)	Algaier & Uyeda (1988)	
PGC-1α	Peroxisome proliferator-activated receptor gamma coactivator 1-alpha	Q9UBK2 (PRGC1_HUMAN)	Knutti, Kaul & Kralli (2000)	
PLC	1-phosphatidylinositol 4,5-bisphosphate phosphodiesterase gamma-1	P19174 (PLCG1_HUMAN)	Rönnstrand (2004)	
PYGL	Glycogen phosphorylase, liver form	P06737 (PYGL_HUMAN)	Zhang et al. (2012)	
SMEK	Serine/threonine-protein phosphatase 4 regulatory subunit 3B
Serine/threonine-protein phosphatase 4 regulatory subunit 3A	Q5MIZ7 (P4R3B_HUMAN)
Q6IN85 (P4R3A_HUMAN)	Chowdhury et al. (2008)	
Glucagon/insulin	
ACC	Acetyl-CoA carboxylase 2
Acetyl-CoA carboxylase 1	O00763 (ACACB_HUMAN)
Q13085 (ACACA_HUMAN)	Cheng et al. (2007) and Moreau et al. (2006)	
AMPK	5′-AMP-activated protein kinase catalytic subunit alpha-2
5′-AMP-activated protein kinase catalytic subunit alpha-1	P54646 (AAPK2_HUMAN)
Q13131 (AAPK1_HUMAN)	Aguan et al. (1994)
Imamura et al. (2001)	
GYS	Glycogen [starch] synthase, muscle
Glycogen [starch] synthase, liver	P13807 (GYS1_HUMAN)
P54840 (GYS2_HUMAN)	Chan et al. (2003)
Bruno et al. (2004)	
PDE3B	cGMP-inhibited 3′,5′-cyclic phosphodiesterase B	Q13370 (PDE3B_HUMAN)	Wilson et al. (2011)	
PHK	Phosphorylase b kinase gamma catalytic chain, liver/testis isoform
Phosphorylase b kinase gamma catalytic chain, skeletal muscle/heart isoform	P15735 (PHKG2_HUMAN)

Q16816 (PHKG1_HUMAN)	Brushia & Walsh (1999)	
PKA	cAMP-dependent protein kinase inhibitor alpha
cAMP-dependent protein kinase catalytic subunit PRKX
cAMP-dependent protein kinase type I-alpha regulatory subunit
cAMP-dependent protein kinase type I-beta regulatory subunit
cAMP-dependent protein kinase catalytic subunit alpha
cAMP-dependent protein kinase type II-alpha regulatory subunit
cAMP-dependent protein kinase catalytic subunit beta
cAMP-dependent protein kinase type II-beta regulatory subunit
cAMP-dependent protein kinase catalytic subunit gamma
cAMP-dependent protein kinase inhibitor beta
cAMP-dependent protein kinase inhibitor gamma	P61925 (IPKA_HUMAN)
P51817 (PRKX_HUMAN)
P10644 (KAP0_HUMAN)

P31321 (KAP1_HUMAN)
P17612 (KAPCA_HUMAN)
P13861 (KAP2_HUMAN)

P22694 (KAPCB_HUMAN)
P31323 (KAP3_HUMAN)
P22612 (KAPCG_HUMAN)
Q9C010 (IPKB_HUMAN)
Q9Y2B9 (IPKG_HUMAN)	Semizarov et al. (1998)
Glesne & Huberman (2006)
Diskar et al. (2010)
Guan, Hou & Ricciardi (2005)
Wang et al. (2000)
Wu et al. (2002)
Mayor et al. (2000)
Miki, Nagashima & Seino (1999)
Dabanaka et al. (2012)
Zhao et al. (2006)	
Glucagon/insulin/somatostatin	
AKT	RAC-alpha serine/threonine-protein kinase
RAC-beta serine/threonine-protein kinase
RAC-gamma serine/threonine-protein kinase	P31749 (AKT1_HUMAN)
P31751 (AKT2_HUMAN)
Q9Y243 (AKT3_HUMAN)	Jones et al. (1991)
Zhang et al. (2006)	
Insulin	
4EBP1	Eukaryotic translation initiation factor 4E-binding protein 1	Q13541 (4EBP1_HUMAN)	Pause et al. (1994)	
aPKC	Protein kinase C alpha type
Protein kinase C iota type
Protein kinase C zeta type	P17252 (KPCA_HUMAN)
P41743 (KPCI_HUMAN)
Q05513 (KPCZ_HUMAN)	Finkenzeller, Marmé & Hug (1990)
Selbie et al. (1993)
Schönwasser et al. (1998)	
BAD*	Bcl2-associated agonist of cell death	Q92934 (BAD_HUMAN)	Wang et al. (1999)	
EiF4E	Eukaryotic translation initiation factor 4E	P06730 (IF4E_HUMAN)	Yanagiya et al. (2012)	
Elk1	ETS domain-containing protein Elk-1	P19419 (ELK1_HUMAN)	Gille et al. (1995)	
FAS	Tumor necrosis factor receptor superfamily member 6	P25445 (TNR6_HUMAN)	Cascino et al. (1995)	
FBP	Fructose-1,6-bisphosphatase 1
Fructose-1,6-bisphosphatase isozyme 2	P09467 (F16P1_HUMAN)
O00757 (F16P2_HUMAN)	El-Maghrabi et al. (1993)
Rakus et al. (2005)	
FOXO1	Forkhead box protein O1	Q12778 (FOXO1_HUMAN)	Shaodong et al. (1999)	
G6PC	Glucose-6-phosphatase catalytic subunit 1
Glucose-6-phosphatase 2
Glucose-6-phosphatase 3	P35575 (G6PC1_HUMAN)
Q9NQR9 (G6PC2_HUMAN)
Q9BUM1 (G6PC3_HUMAN)	Pan et al. (1998)
Petrolonis et al. (2004)
Martin et al. (2002)	
GK	Glycerol kinase
Glycerol kinase 2
Glycerol kinase 3	P32189 (GLPK_HUMAN)
Q14410 (GLPK2_HUMAN)
Q14409 (GLPK3_HUMAN)	Stepanian et al. (2003)
Chen et al. (2017)
Ohira et al. (2005)	
GLUT1	Solute carrier family 2, facilitated glucose transporter member 1	P11166 (GTR1_HUMAN)	Mueckler & Makepeace (2008)	
GLUT4	Solute carrier family 2, facilitated glucose transporter member 4	P14672 (GLUT4_HUMAN)	Kawanishi et al. (2000)	
GRB2	Growth factor receptor-bound protein 2	P62993 (GRB2_HUMAN)	Lowenstein et al. (1992)	
GSK-3	Glycogen synthase kinase-3 alpha
Glycogen synthase kinase-3 beta	P49840 (GSK3A_HUMAN)
P49841 (GSK3B_HUMAN)	Nikoulina et al. (2000)
Boyle et al. (1991)	
HSL	Hormone-sensitive lipase	Q05469 (LIPS_HUMAN)	Holst et al. (1996)	
INS	Insulin	P01308 (INS_HUMAN)	Bremser et al. (1999)	
INSR	Insulin receptor	P06213 (INSR_HUMAN)	Xu, Paxton & Fujita-Yamaguchi (1990)	
IRS	Insulin receptor substrate 1
Insulin receptor substrate 2
Insulin receptor substrate 4	P35568 (IRS1_HUMAN)
Q9Y4H2 (IRS2_HUMAN)
O14654 (IRS4_HUMAN)	Kuo et al. (2007)
Watowich et al. (1999)
Fantin et al. (1998)	
MNK	MAP kinase-interacting serine/threonine-protein kinase 1
MAP kinase-interacting serine/threonine-protein kinase 2	Q9BUB5 (MKNK1_HUMAN)
Q9HBH9 (MKNK2_HUMAN)	Fukunaga & Hunter (1997)
Scheper et al. (2001)	
mTOR	Serine/threonine-protein kinase mTOR	P42345 (MTOR_HUMAN)	Kim et al. (2002)	
p70S6K	Ribosomal protein S6 kinase beta-1
Ribosomal protein S6 kinase beta-2	P23443 (KS6B1_HUMAN)
Q9UBS0 (KS6B2_HUMAN)	Pullen et al. (1998)
Nguyen et al. (2018)	
PDK1/2	3-phosphoinositide-dependent protein kinase 1
[Pyruvate dehydrogenase (acetyl-transferring)] kinase isozyme 2, mitochondrial	O15530 (PDPK1_HUMAN)
Q15119 (PDK2_HUMAN)	Alessi et al. (1997)
Gudi et al. (1995)	
PEPCK	Phosphoenolpyruvate carboxykinase, cytosolic [GTP]
Phosphoenolpyruvate carboxykinase [GTP], mitochondrial	P35558 (PCKGC_HUMAN)
Q16822 (PCKGM_HUMAN)	Zhao et al. (2010)
Lu et al. (2008)	
PIP3	Phosphatidylinositol 3,4,5-trisphosphate 3-phosphatase and dual-specificity protein phosphatase PTEN	P60484 (PTEN_HUMAN)	Li & Sun (1997)	
PPI	Protein phosphatase inhibitor 2	P41236 (IPP2_HUMAN)	Sakashita et al. (2003)	
PYG	Pygopus homolog 1
Pygopus homolog 2	Q9Y3Y4 (PYGO1_HUMAN)
Q9BRQ0 (PYGO2_HUMAN)	Fiedler et al. (2008)
Thompson et al. (2002)	
PYK	Protein-tyrosine kinase 2-beta	Q14289 (FAK2_HUMAN)	Lev et al. (1995)	
Raptor	Regulatory-associated protein of mTOR	Q8N122 (RPTOR_HUMAN)	Kim et al. (2002)	
RAS	GTPase Eras
GTPase HRas
GTPase KRas
GTPase NRas	Q7Z444 (RASE_HUMAN)
P01112 (RASH_HUMAN)
P01116 (RASK_HUMAN)
P01111 (RASN_HUMAN)	Zhang et al. (2010)
Guil et al. (2003)
Yang et al. (2012)
Yin et al. (2019)	
Rheb	GTP-binding protein Rheb	Q15382 (RHEB_HUMAN)	Tee et al. (2002)	
S6	Ribosomal protein S6 kinase alpha-1
Ribosomal protein S6 kinase alpha-2
Ribosomal protein S6 kinase alpha-3
Ribosomal protein S6 kinase alpha-4
Ribosomal protein S6 kinase alpha-5
Ribosomal protein S6 kinase alpha-6
Ribosomal protein S6 kinase beta-1
Ribosomal protein S6 kinase beta-2
Ribosomal protein S6 kinase delta-1	Q15418 (KS6A1_HUMAN)
Q15349 (KS6A2_HUMAN)
P51812 (KS6A3_HUMAN)
O75676 (KS6A4_HUMAN)
O75582 (KS6A5_HUMAN)
Q9UK32 (KS6A6_HUMAN)
P23443 (KS6B1_HUMAN)
Q9UBS0 (KS6B2_HUMAN)
Q96S38 (KS6C1_HUMAN)	Dalby et al. (1998)
Zhao et al. (1995)
Sutherland, Leighton & Cohen (1993)
Pierrat et al. (1998)
Deak et al. (1998)
Berns et al. (2004)
Pullen et al. (1998)
Nguyen et al. (2018)
Hayashi et al. (2002)	
SHC	SHC-transforming protein 1
SHC-transforming protein 2
SHC-transforming protein 3
SHC-transforming protein 4	P29353 (SHC1_HUMAN)
P98077 (SHC2_HUMAN)
Q92529 (SHC3_HUMAN)
Q6S5L8 (SHC4_HUMAN)	Rönnstrand (2004)
Warner et al. (2000)
Hennige et al. (2000)
Fagiani et al. (2007)	
SHIP2	Phosphatidylinositol 3,4,5-trisphosphate 5-phosphatase 2	O15357 (SHIP2_HUMAN)	Habib et al. (1998)	
SOS	Son of sevenless homolog 1
Son of sevenless homolog 2	Q07889 (SOS1_HUMAN)
Q07890 (SOS2_HUMAN)	Chardin et al. (1993)
Umikawa et al. (1999)	
SREBP-1c	Sterol regulatory element-binding protein 1	P36956 (SRBP1_HUMAN)	Yokoyama et al. (1993)	
TSC1	Hamartin	Q92574 (TSC1_HUMAN)	Tee et al. (2002)	
TSC2	Tuberin	P49815 (TSC2_HUMAN)	
Insulin/somatostatin	
ERK1/2	Mitogen-activated protein kinase 3
Mitogen-activated protein kinase 1	P27361 (MK03_HUMAN)
P28482 (MK01_HUMAN)	Marklund et al. (1993)
Sgouras et al. (1995)	
JNK	Mitogen-activated protein kinase 8
Mitogen-activated protein kinase 9
Mitogen-activated protein kinase 10	P45983 (MK08_HUMAN)
P45984 (MK09_HUMAN)
P53779 (MK10_HUMAN)	Gupta et al. (1996)
Sluss et al. (1994)
Lisnock et al. (2000)	
MEK1/2	Dual specificity mitogen-activated protein kinase kinase 1
Dual specificity mitogen-activated protein kinase kinase 2	Q02750 (MP2K1_HUMAN)
P36507 (MP2K2_HUMAN)	Liu et al. (2004)
Mittal, Peak-Chew & McMahon (2006)	
P13K	Phosphatidylinositol 3-kinase regulatory subunit alpha	P27986 (P85A_HUMAN)	Vainikka et al. (1994)	
RAF	RAF proto-oncogene serine/threonine-protein kinase	P04049 (RAF1_HUMAN)	Dubois et al. (1997)	
Somatostatin	
BAX	Apoptosis regulator BAX	Q07812 (BAX_HUMAN)	Oltval, Milliman & Korsmeyer (1993)	
NF-κB	Nuclear factor NF-kappa-B p105 subunit
Nuclear factor NF-kappa-B p100 subunit	P19838 (NFKB1_HUMAN)
Q00653 (NFKB2_HUMAN)	Beinke et al. (2004)
Dobrzanski, Ryseck & Bravo (1994)	
P21	Cyclin-dependent kinase inhibitor 1	P38936 (CDN1A_HUMAN)	Harper et al. (1993)	
P27	Cyclin-dependent kinase inhibitor 1B	P46527 (CDN1B_HUMAN)	Ishida et al. (2000)	
P53	Cellular tumor antigen p53	P04637 (P53_HUMAN)	Schneider, Montenarh & Wagner (1998)	
SHP1	Tyrosine-protein phosphatase non-receptor type 6	P29350 (PTN6_HUMAN)	Keilhack et al. (2001)	
SHP2	Tyrosine-protein phosphatase non-receptor type 11	Q06124 (PTN11_HUMAN)	Miao et al. (2000)	
SST*	Somatostatin	P61278 (SMS_HUMAN)	Luque & Kineman (2018)	
SSTR	Somatostatin receptor type 1
Somatostatin receptor type 2
Somatostatin receptor type 3
Somatostatin receptor type 4
Somatostatin receptor type 5	P30872 (SSR1_HUMAN)
P30874 (SSR2_HUMAN)
P32745 (SSR3_HUMAN)
P31391 (SSR4_HUMAN)
P35346 (SSR5_HUMAN)	Pasquali et al. (2001)
Grant, Collier & Kumar (2004)
Yamada et al. (1992)
Panetta et al. (1994)	
Zac1	Zinc finger protein PLAGL1	Q9UM63 (PLAL1_HUMAN)	Kas et al. (1998)	
Note:

* No suitable pdam homolog found.

Table 2 Remote homology detection of candidate insulin signaling related proteins in Pocillopora damicornis.

Full list of proteins involved in INS and INS-related signaling pathways (i.e., GCG and SST signaling), matched to Pdam ID with number of residues overlayed (Cols), P-value, E-value, matched sequence length, probability, query template length and percentage (%) identity retrieved from hhblits.

Gene	Uniport ID	Protein name	Match columns	Pdam ID	Prob	E-value	P-value	Aligned Cols	Query HMM	Template HMM	% Identity	
Glucagon	
ADCY8	P40145	Adenylate cyclase type 8	1,251	pdam_00002623	100	2.80E−94	7.20E–98	983	155–1,179	109–1,111 (1,122)	36	
CALM	P0DP23; P0DP24; P0DP25	Calmodulin-1; Calmodulin-2; Calmodulin-3	149	pdam_00003911	100	6.70E−42	1.70E–45	143	3–147	75–249 (921)	20	
CBP	Q92793	CREB-binding protein	2,442	pdam_00013067	100	3E−319	7E–323	2,061	1–2,442	1–2,199 (2,199)	47	
CREB1	P16220	Cyclic AMP-responsive element-binding protein 1	327	pdam_00005762	100	7.20E−67	1.30E–70	252	63–327	39–319 (319)	52	
CRTC2	Q53ET0	CREB-regulated transcription coactivator 2	693	pdam_00014061	100	2.90E−82	5.10E–86	460	17–693	2–506 (506)	34	
GCG *	P01275	Pro-glucagon	180	pdam_00011985	13	15	0.0022	12	130–141	5–16 (74)	50	
GCGR	P47871	Glucagon receptor	477	pdam_00008152	100	1.90E−49	4.00E–53	371	25–431	23–410 (765)	22	
GLP-1R	P43220	Glucagon-like peptide 1 receptor	463	pdam_00008152	100	1.60E−43	3.30E–47	377	13–424	12–401 (765)	22	
GNAQ	P50148	Guanine nucleotide-binding protein G(q) subunit alpha	359	pdam_00011071	100	7.50E−77	1.90E–80	352	7–359	1–365 (365)	51	
GNAS	P63092	Guanine nucleotide-binding protein G(s) subunit alpha isoforms short	394	pdam_00011071	100	3.30E−61	8.20E–65	352	1–393	1–364 (365)	44	
Q5JWF2	Guanine nucleotide-binding protein G(s) subunit alpha isoforms XLas	1,037	pdam_00011071	100	4.70E−56	1.20E–59	342	662–1,037	12–365 (365)	44	
O95467	Neuroendocrine secretory protein 55	245	pdam_00011481	67.4	0.35	5.00E−05	23	33–55	5–27 (973)	26	
IP3R	Q14573	Inositol 1,4,5-trisphosphate receptor type 3	2,671	pdam_00007499	100	0.00E+00	0.00E+00	2,511	1–2,668	1–2,653 (2,667)	58	
PFKFB1	P16118	6-phosphofructo-2-kinase/fructose-2,6-bisphosphatase 1	471	pdam_00000774	100	1.60E−69	4.00E−73	424	47–471	1–425 (425)	60	
PGC-1α	Q9UBK2	Peroxisome proliferator-activated receptor gamma coactivator 1-alpha	798	pdam_00012386	99.7	6.90E−22	1.20E−25	134	663–798	413–548 (559)	40	
PLC	P19174	1-phosphatidylinositol 4,5-bisphosphate phosphodiesterase gamma-1	1,290	pdam_00015403	100	2.00E−170	3.00E−174	1,116	24–1,219	19–1,177 (1,293)	48	
PYGL	P06737	Glycogen phosphorylase, liver form	847	pdam_00018058	100	3.00E−186	5.00E−190	814	1–834	2–818 (826)	68	
SMEK	Q5MIZ7	Serine/threonine-protein phosphatase 4 regulatory subunit 3B	849	pdam_00021014	100	2.00E−160	4.00E−164	676	1–725	1–683 (753)	62	
Q6IN85	Serine/threonine-protein phosphatase 4 regulatory subunit 3A	833	pdam_00021014	100	6.00E−158	1.00E−161	676	1–692	1–684 (753)	63	
Glucagon/insulin	
ACC	O00763	Acetyl-CoA carboxylase 2	2,458	pdam_00001946	100	0.00E+00	0.00E+00	2,163	240–2,458	26–2,210 (2,219)	62	
	Q13085	Acetyl-CoA carboxylase 1	2,346	pdam_00001946	100	0.00E+00	0.00E+00	2,167	96–2,346	24–2,209 (2,219)	64	
AMPK	P54646	5′-AMP-activated protein kinase catalytic subunit alpha-2	552	pdam_00023206	100	6.00E−106	1.00E−109	518	9–551	9–551	62	
Q13131	5′-AMP-activated protein kinase catalytic subunit alpha-1	559	pdam_00023206	100	5.00E−100	1.00E−103	517	19–558	35–568 (569)	62	
GYS	P13807	Glycogen [starch] synthase, muscle	737	pdam_00011808	100	5.60E−98	1.00E−101	640	3–662	80–720 (727)	59	
P54840	Glycogen [starch] synthase, liver	703	pdam_00011808	100	2.30E−99	5.00E−103	640	3–662	80–720 (727)	59	
PDE3B	Q13370	cGMP-inhibited 3′,5′-cyclic phosphodiesterase 3B	1,112	pdam_00001873	100	1.00E−165	2.00E−169	926	10–1,090	5–984 (1,033)	36	
PHK	P15735	Phosphorylase b kinase gamma catalytic chain, liver/testis isoform	406	pdam_00008094	100	2.40E−77	5.70E−81	368	7–379	6–376 (420)	53	
Q16816	Phosphorylase b kinase gamma catalytic chain, skeletal muscle/heart isoform	387	pdam_00008094	100	7.60E−83	1.80E−86	370	5–381	8–381 (420)	53	
PKA	P61925	cAMP-dependent protein kinase inhibitor alpha	76	pdam_00011290	99.4	3.10E−18	5.20E−22	56	1–56	1–57 (70)	30	
P51817	cAMP-dependent protein kinase catalytic subunit PRKX	358	pdam_00008088	100	3.40E−56	7.10E−60	315	43–358	202–523 (699)	31	
P10644	cAMP-dependent protein kinase type I-alpha regulatory subunit	381	pdam_00010058	100	4.00E−61	1.00E−64	367	8–380	6–372 (373)	70	
P31321	cAMP-dependent protein kinase type I-beta regulatory subunit	381	pdam_00010058	100	4.30E−59	1.10E−62	363	13–381	11–373 (373)	70	
P17612	cAMP-dependent protein kinase catalytic subunit alpha	351	pdam_00024226	100	2.40E−77	5.70E−81	346	6–351	3–348 (348)	82	
P13861	cAMP-dependent protein kinase type II-alpha regulatory subunit	404	pdam_00018522	100	5.10E−63	1.20E−66	381	2–396	59–440 (444)	54	
P22694	cAMP-dependent protein kinase catalytic subunit beta	351	pdam_00024226	100	4.80E−76	1.20E−79	344	8–351	5–348 (348)	82	
P31323	cAMP-dependent protein kinase type II-beta regulatory subunit	418	pdam_00018522	100	3.90E−61	9.40E−65	378	2–407	60–437 (444)	54	
P22612	cAMP-dependent protein kinase catalytic subunit gamma	351	pdam_00024226	100	6.30E−78	1.50E−81	340	12–351	9–348 (348)	75	
Q9C010	cAMP-dependent protein kinase inhibitor beta	78	pdam_00011290	99.3	7.50E−17	1.30E−20	59	8–66	1–60 (70)	36	
Q9Y2B9	cAMP-dependent protein kinase inhibitor gamma	76	pdam_00011290	99.7	3.00E−23	5.10E−27	52	1–52	1–53 (70)	31	
Glucagon/insulin/somatostatin	
AKT	P31749	RAC-alpha serine/threonine-protein kinase	480	pdam_00008116	100	2.40E−70	5.20E−74	331	141–477	55–389 (500)	45	
P31751	RAC-beta serine/threonine-protein kinase	481	pdam_00008116	100	1.80E−71	4.00E−75	330	144–478	56–389 (500)	46	
Q9Y243	RAC-gamma serine/threonine-protein kinase	479	pdam_00001122	100	2.20E−71	5.00E−65	400	22–441	104–516 (1,524)	25	
4EBP1	Q13541	Eukaryotic translation initiation factor 4E-binding protein 1	118	pdam_00003235	100	2.10E−39	3.50E−43	116	1–118	1–119 (119)	55	
aPKC	P17252	Protein kinase C alpha type	672	pdam_00020998	100	8.00E−134	2.00E−137	648	18–665	10–711 (724)	66	
P41743	Protein kinase C iota type	596	pdam_00005220	100	1.00E−139	2.00E−143	573	24–596	8–692 (692)	71	
Q05513	Protein kinase C zeta type	592	pdam_00005220	100	5.00E−124	1.00E−127	578	9–592	3–692 (692)	66	
BAD *	Q92934	Bcl2-associated agonist of cell death	168	pdam_00004189	5.3	34	0.0073	42	119–160	24–65 (293)	12	
EiF4E	P06730	Eukaryotic translation initiation factor 4E	217	pdam_00019745	100	1.10E−43	2.50E−47	186	32–217	37–232 (232)	63	
Elk1	P19419	ETS domain-containing protein Elk-1	428	pdam_00014011	100	6.60E−39	1.30E−42	163	4–177	14–181 (204)	41	
FAS	P25445	Tumor necrosis factor receptor superfamily member 6	335	pdam_00022994	99.9	4.20E−28	8.90E−32	242	60–317	44–399 (409)	17	
FBP	P09467	Fructose-1,6-bisphosphatase 1	338	pdam_00021399	100	1.80E−41	4.10E−45	239	7–332	4–244 (246)	61	
O00757	Fructose-1,6-bisphosphatase isozyme 2	339	pdam_00021399	100	2.40E−40	5.50E−44	240	6–332	3–244 (246)	57	
FOXO1	Q12778	Forkhead box protein O1	655	pdam_0001017	100	6.90E−96	1.30E−99	434	154–639	50–546 (557)	37	
G6PC	P35575	Glucose-6-phosphatase catalytic subunit 1	357	pdam_00002925	100	4.40E−43	1.10E−46	315	5–348	1–324 (505)	29	
Q9NQR9	Glucose-6-phosphatase 2 OS=Homo sapiens	355	pdam_00002925	100	1.00E−43	2.40E−47	318	1–350	1–328 (505)	33	
Q9BUM1	Glucose-6-phosphatase 3	346	pdam_00002925	100	7.60E−42	1.80E−45	315	1–338	1–327 (505)	31	
GK	P32189	Glycerol kinase OS=Homo sapiens	559	pdam_00006960	100	2.00E−47	4.90E−51	427	11–474	8–476 (476)	19	
Q14410	Glycerol kinase 2 OS=Homo sapiens	553	pdam_00002235	100	6.50E−46	1.60E−49	470	12–507	5–547 (559)	23	
Q14409	Glycerol kinase 3	553	pdam_00006960	100	1.00E−46	2.50E−50	424	11–468	8–476 (476)	18	
GLUT1	P11166	Solute carrier family 2, facilitated glucose transporter member 1	492	pdam_00006372	100	1.40E−40	4.10E−44	445	15–470	40–484 (495)	45	
GLUT4	P14672	Solute carrier family 2, facilitated glucose transporter member 4	509	pdam_00014912	100	3.60E−39	9.90E−43	437	23–485	27–484 (494)	28	
GRB2	P62993	Growth factor receptor-bound protein 2	217	pdam_00010914	100	3.50E−36	1.00E−39	213	1–213	1–217 (218)	63	
GSK-3	P49840	Glycogen synthase kinase-3 alpha	483	pdam_00001353	100	2.60E−75	6.70E−79	359	89–447	23–388 (421)	81	
P49841	Glycogen synthase kinase-3 beta	420	pdam_00001353	100	2.00E−80	5.00E−84	381	1–383	1–387 (421)	84	
HSL	Q05469	Hormone-sensitive lipase	1,076	pdam_00001853	100	2.40E−96	5.00E−100	709	304–1,058	19–795 (803)	38	
INS	P01308	Insulin	110	pdam_00006633	98.8	3.20E−13	6.90E−17	101	2–109	5–116 (116)	24	
INSR	P06213	Insulin receptor	1,382	pdam_00013976	100	1.00E−187	2.00E−191	1,164	28–1,307	8–1,274 (1,306)	42	
IRS	P35568	Insulin receptor substrate 1	1,242	pdam_00006434	100	2.40E−36	3.90E−40	268	6–275	8–312 (1,427)	21	
Q9Y4H2	Insulin receptor substrate 2	1,338	pdam_00006434	99.9	4.90E−34	7.80E−38	292	25–329	9–337 (1,427)	21	
O14654	Insulin receptor substrate 4	1,257	pdam_00015468	99.8	1.90E−26	3.30E−30	217	74–334	9–225 (797)	39	
MNK	Q9BUB5	MAP kinase-interacting serine/threonine-protein kinase 1	465	pdam_00010796	100	8.90E−49	2.10E−52	369	37–451	99–472 (490)	57	
Q9HBH9	MAP kinase-interacting serine/threonine-protein kinase 2	465	pdam_00010796	100	3.00E−52	7.20E−56	374	73–454	100–475 (490)	58	
mTOR	P42345	Serine/threonine-protein kinase mTOR	2,549	pdam_00009963	100	1.00E−280	4.00E−284	2,353	21–2,548	1–2,369 (2,401)	68	
p70S6K	P23443	Ribosomal protein S6 kinase beta-1	525	pdam_00008116	100	7.80E−87	1.70E−90	421	24–454	1–425 (500)	71	
Q9UBS0	Ribosomal protein S6 kinase beta-2	482	pdam_00008116	100	5.30E−85	1.20E−88	376	48–427	45–422 (500)	70	
PDK1/2	O15530	3-phosphoinositide-dependent protein kinase 1	556	pdam_00003014-	100	1.20E−78	2.90E−82	302	175–549	54–356 (360)	65	
Q15119	[Pyruvate dehydrogenase (acetyl-transferring)] kinase isozyme 2, mitochondrial	407	pdam_0001703	100	2.10E−51	5.30E−55	390	1–398	1–394 (422)	50	
PEPCK	P35558	Phosphoenolpyruvate carboxykinase, cytosolic [GTP]	622	pdam_00016705	100	1.00E−187	2.00E−191	605	12–621	41–647 (647)	63	
Q16822	Phosphoenolpyruvate carboxykinase [GTP], mitochondrial	640	pdam_00016705	100	1.00E−188	2.00E−192	607	28–639	40–647 (647)	62	
PIP3	P60484	Phosphatidylinositol 3,4,5-trisphosphate 3-phosphatase and dual-specificity protein phosphatase PTEN	403	pdam_00003212	100	4.50E−60	1.00E−63	340	2–354	18–357 (436)	52	
PPI	P41236	Protein phosphatase inhibitor 2	205	pdam_00000730	100	2.50E−38	4.40E−42	142	6–199	4–147 (159)	39	
PYG	Q9Y3Y4	Pygopus homolog 1	419	pdam_00023740	99.2	1.20E−15	2.40E−19	68	332–400	55–124 (345)	28	
	Q9BRQ0	Pygopus homolog 2	406	pdam_00023740	99.2	9.60E−16	1.70E−19	93	301–394	74–167 (186)	22	
PYK	Q14289	Protein-tyrosine kinase 2-beta	1,009	pdam_00008953	100	1.00E−144	2.00E−148	954	36–1,002	20–1,118 (1,124)	42	
Raptor	Q8N122	Regulatory-associated protein of mTOR	1,335	pdam_00003007	100	8.00E−245	2.00E−248	1,295	12–1,332	11–1,333 (1,338)	64	
RAS	Q7Z444	RASE_HUMAN GTPase ERas	233	pdam_00018296	99.8	1.30E−26	3.40E−30	165	37–201	11–195 (253)	27	
P01112	GTPase HRas	189	pdam_00015181	100	5.30E−35	1.60E−38	180	1–181	1–180 (183)	83	
P01116	GTPase KRas OS=Homo sapiens	189	pdam_00015181	99.9	1.40E−33	4.30E−37	183	1–189	1–183 (183)	84	
P01111	GTPase NRas	189	pdam_00015181	99.9	4.10E−34	1.20E−37	174	1–175	1–174 (183)	84	
Rheb	Q15382	GTP-binding protein Rheb	184	pdam_00010612	99.9	4.80E−29	1.50E−32	184	1–184	1–185 (185)	65	
S6	Q15418	Ribosomal protein S6 kinase alpha-1	735	pdam_00004023	100	5.40E−98	1.00E−101	591	55–730	110–707 (710)	67	
Q15349	Ribosomal protein S6 kinase alpha-2	733	pdam_00004023	100	9.00E−100	2.00E−103	589	52–725	110–705 (710)	69	
P51812	Ribosomal protein S6 kinase alpha-3	740	pdam_00004023	100	1.60E−98	4.00E−102	588	60–731	109–704 (710)	69	
O75676	Ribosomal protein S6 kinase alpha-4	772	pdam_00021026	100	2.50E−99	6.00E−103	697	23–733	24–728 (757)	60	
O75582	Ribosomal protein S6 kinase alpha-5	802	pdam_00021026	100	2.00E−110	5.00E−114	706	30–745	14–727 (757)	66	
Q9UK32	Ribosomal protein S6 kinase alpha-6	745	pdam_00021026	100	6.40E−95	1.70E−98	635	54–692	15–676 (757)	44	
Q96S38	Ribosomal protein S6 kinase delta-1	1,066	pdam_00020764	99.9	3.20E−33	6.10E−37	167	900–1,066	789–955 (958)	51	
SHC	P29353	SHC-transforming protein 1	583	pdam_00010704	100	2.10E−84	4.50E−88	441	130–582	10–550 (550)	41	
P98077	SHC-transforming protein 2	582	pdam_00010704	100	1.30E−77	2.60E−81	448	122–580	11–549 (550)	38	
Q92529	SHC-transforming protein 3	594	pdam_00010704	100	1.60E−74	3.40E−78	443	133–593	20–550 (550)	37	
Q6S5L8	SHC-transforming protein 4	630	pdam_00010704	100	1.50E−68	3.20E−72	433	183–619	33–549 (550)	41	
SHIP2	O15357	Phosphatidylinositol 3,4,5-trisphosphate 5-phosphatase 2	1,258	pdam_00012957	100	3.00E−124	6.00E−128	735	18–897	2–746 (873)	34	
SOS	Q07889	Son of sevenless homolog 1	1,333	pdam_00007801	100	2.00E−191	4.00E−195	1,061	8–1,076	4–1,079 (1,252)	56	
Q07890	Son of sevenless homolog 2	1,332	pdam_00007801	100	1.00E−192	3.00E−196	1,077	8–1,095	4–1,094 (1,252)	56	
SREBP-1c	P36956	Sterol regulatory element-binding protein 1	1,147	pdam_00001678	100	3.00E−166	5.00E−170	822	292–1,144	411–1,296 (1,297)	42	
TSC1	Q92574	Hamartin	1,164	pdam_00004350	100	6.00E−144	1.00E−147	720	7–771	12–845 (857)	34	
TSC2	P49815	Tuberin	1,807	pdam_00021002	100	2.00E−216	4.00E−220	1,595	3–1,807	2–1,826 (1,828)	39	
Insulin/somatostatin	
ERK1/2	P27361	Mitogen-activated protein kinase 3	379	pdam_00020223	100	2.90E−74	7.50E−78	346	29–374	40–385 (396)	81	
P28482	Mitogen-activated protein kinase 1	360	pdam_00020223	100	1.20E−75	3.00E−79	350	7–356	35–384 (396)	84	
JNK	P45983	Mitogen-activated protein kinase 8	427	pdam_00012090	100	1.60E−61	3.80E−65	364	24–407	35–424 (426)	37	
P45984	Mitogen-activated protein kinase 9	424	pdam_00012090	100	1.90E−61	4.70E−65	330	24–364	35–390 (426)	39	
P53779	Mitogen-activated protein kinase 10	464	pdam_00012090	100	2.60E−61	6.40E−65	330	62–402	35–390 (426)	39	
MEK1/2	Q02750	Dual specificity mitogen-activated protein kinase kinase 1	393	pdam_00000776	100	1.10E−60	2.90E−64	362	3–374	8–373 (373)	69	
P36507	Dual specificity mitogen-activated protein kinase kinase 2	400	pdam_00000776	100	9.10E−58	2.30E−61	359	10–382	12–373 (373)	69	
P13K	P27986	Phosphatidylinositol 3-kinase regulatory subunit alpha	724	pdam_00005911	100	1.10E−69	2.70E−73	532	176–721	344–889 (894)	38	
RAF	P04049	RAF proto-oncogene serine/threonine-protein kinase	648	pdam_00015444	100	5.90E−46	1.40E−49	246	342–599	377–628 (715)	23	
Somatostatin	
BAX	Q07812	Apoptosis regulator BAX	192	pdam_00002763	99.9	2.90E−33	5.90E−37	177	13–190	47–250 (251)	27	
NF-κB	P19838	Nuclear factor NF-kappa-B p105 subunit	968	pdam_00003205	100	7.00E−113	1.00E−116	790	39–891	40–860 (926)	46	
P21	P38936	Cyclin-dependent kinase inhibitor 1	164	pdam_00006472	99.3	1.90E−16	3.40E−20	70	13–82	22–95 (197)	33	
P27	P46527	Cyclin-dependent kinase inhibitor 1B	198	pdam_00006472	99.4	1.60E−17	2.90E−21	83	13–95	11–97 (197)	33	
P53	P04637	Cellular tumor antigen p53	393	pdam_00016598	100	2.20E−66	4.00E−70	248	95–352	144–397 (439)	37	
SHP1	P29350	Tyrosine-protein phosphatase non-receptor type 6	595	pdam_00004172	100	2.00E−86	5.20E−90	508	3–523	5–513 (590)	59	
SHP2	Q06124	Tyrosine-protein phosphatase non-receptor type 11	593	pdam_00004172	100	1.10E−94	2.70E−98	515	2–533	2–517 (590)	65	
SST *	P61278	Somatostatin	116	pdam_00016702	12.1	13	0.0024	50	28–77	131–197 (325)	28	
SSTR	P30872	Somatostatin receptor type 1	391	pdam_00013387	100	2.10E−44	5.70E−48	287	57–345	214–520 (773)	18	
P30874	Somatostatin receptor type 2	369	pdam_00012380	100	1.20E−46	3.20E−50	304	27–332	199–518 (773)	19	
P32745	Somatostatin receptor type 3	418	pdam_00019641	100	3.20E−53	7.20E−57	364	45–412	132–573 (976)	19	
P31391	Somatostatin receptor type 4	388	pdam_00004106	100	4.80E−42	1.40E−45	295	39–334	35–342 (380)	21	
P35346	Somatostatin receptor type 5	364	pdam_00020795	100	6.40E−44	1.80E−47	290	34–325	28–360 (425)	27	
Zac1	Q9UM63	Zinc finger protein PLAGL1	463	pdam_00020164	100	1.70E−34	4.70E−38	218	3–220	114–374 (479)	20	
Note:

* No suitable pdam homolog found (bolded rows); E-value refers to the expected number of random hits for a given alignment score; % Identity is the number of identical residues.

Overview of the INS signaling pathway. The INSR is a disulfide-linked tetramer that contains two α and two β subunits located on the cell surface (Xu, Paxton & Fujita-Yamaguchi, 1990; Bremser et al., 1999; Watowich et al., 1999; Kuo et al., 2007; Kumar, Vizgaudis & Klein-Seetharaman, 2021). The α subunits contain four INS binding sites in the extracellular domain (Xu, Paxton & Fujita-Yamaguchi, 1990; Bremser et al., 1999; Kumar, Vizgaudis & Klein-Seetharaman, 2021). The β subunits contain the intracellular tyrosine kinase domains (Xu, Paxton & Fujita-Yamaguchi, 1990; Kumar, Vizgaudis & Klein-Seetharaman, 2021). INS binding to the α subunits activates tyrosine kinase by changing conformations of the β subunits (Xu, Paxton & Fujita-Yamaguchi, 1990; Bremser et al., 1999; Kuo et al., 2007). This causes phosphorylation of insulin receptor substrates (IRS) and SHC which activates kinase cascades such as the PI3K/AKT and the MAPK signaling pathways (Xu, Paxton & Fujita-Yamaguchi, 1990; Jones et al., 1991; Vainikka et al., 1994; Fukunaga & Hunter, 1997; Watowich et al., 1999; Hennige et al., 2000; Zhang et al., 2006; Kuo et al., 2007). IRS activates PI3K to produce the secondary messenger PI-(3,4,5)trisphosphate (PIP3) by accelerating phosphatidylinositol (PI) phosphorylation across the membrane (Vainikka et al., 1994; Li & Sun, 1997; Fantin et al., 1998). IRS-1 integrates signals from different pathways and is considered the master regulator of INS sensitivity (Xu, Paxton & Fujita-Yamaguchi, 1990; Bremser et al., 1999; Kuo et al., 2007). SHIP2 acts as a PI3K antagonist and is involved in the downregulation of the AKT pathway by PIP3 degradation (Jones et al., 1991; Chardin et al., 1993; Habib et al., 1998; Zhang et al., 2006). PDK1/2 is recruited by PIP3 and phosphorylates AKT on serine/threonine residues (Li & Sun, 1997; Fukunaga & Hunter, 1997; Fantin et al., 1998). Subsequently, AKT phosphorylation activates several substrates including FOXO1 proteins that regulate TCF for gene transcription and GSK3 which regulates GLUT4 translocation to the cell membrane (Boyle et al., 1991; Holst et al., 1996; Shaodong et al., 1999; Nikoulina et al., 2000). Activation of IRS-1 also binds to GRB2 and SOS to activate RAS (Lowenstein et al., 1992; Chardin et al., 1993; Umikawa et al., 1999; Guil et al., 2003; Zhang et al., 2010). Activated SHC catalyzes GTP exchange by activating RAS on the plasma membrane (Warner et al., 2000; Hennige et al., 2000; Guil et al., 2003; Rönnstrand, 2004; Zhang et al., 2010; Yin et al., 2019); RAS activation leads to MAPK cascade activation via ERK phosphorylation (Sgouras et al., 1995; Gupta et al., 1996; Tee et al., 2002; Yang et al., 2012; Yin et al., 2019). The AKT pathway also mediates cell survival including growth, differentiation, and inhibiting apoptosis. INS synthesis at the translational level is mediated by mTOR which phosphorylates INS and promotes INS synthesis (Dalby et al., 1998; Kim et al., 2002; Tee et al., 2002). On the other hand, phosphorylation of serine residues on IRS-1 by its downstream effector protein kinase C-ζ (aPKC) which downregulates INS signaling and impairs P13K activity, constitutes negative feedback mechanism in response to INS (Vainikka et al., 1994; Kim et al., 2002; Kuo et al., 2007; Lee et al., 2008). In response to stimuli such as growth factors, IRS-1 phosphorylates multiple serine/threonine kinases such as p70 S6 kinase (P70S6K), mTOR and JNK (Sluss et al., 1994; Pullen et al., 1998; Deak et al., 1998; Watowich et al., 1999; Berns et al., 2004; Kuo et al., 2007; Nguyen et al., 2018).

Overview of GCG signaling pathway. As described above, glucose levels in the body are largely dependent on the coordinated release of GCG by α-cells and INS by β-cells of pancreatic islet, respectively, as shown in Fig. 1 (Orskov, Wettergren & Holst, 1993; MacNeil et al., 1994; Quesada et al., 2008). During hyperglycemia, INS secretion from β-cells is induced; however, during hypoglycemia α-cell secretion of GCG is induced. GCG exerts action via activation of GCG signaling and ADCY8 by coupling to GPCRs such as GCGR and GLP-1R (Orskov, Wettergren & Holst, 1993; MacNeil et al., 1994; Wang, Liang & Wang, 2013). The effects of INS are thus counterbalanced by GCG and the GCG signaling pathway promotes glucose production (Orskov, Wettergren & Holst, 1993; MacNeil et al., 1994; Quesada et al., 2008). GCG binding activates the receptors GNAS and GNAQ of G protein-mediated signaling (Orskov, Wettergren & Holst, 1993; Montrose-Rafizadeh et al., 1999; O’Donovan et al., 1999; Pak, Pham & Rotin, 2002; Alvarez-Curto et al., 2016). GNAS activates the PKA pathway and cAMP production (Orskov, Wettergren & Holst, 1993; MacNeil et al., 1994; Montrose-Rafizadeh et al., 1999). An increase in intracellular cAMP production phosphorylates CREB transcription factor to increase gluconeogenesis (Zhang & Bieker, 1998; O’Donovan et al., 1999). PKA activation also results in the inhibition of glycolysis by the inactivation of PFKFB which is one of the key enzymes in glucose metabolism (Algaier & Uyeda, 1988; Miki, Nagashima & Seino, 1999; Wang et al., 2000; Wu et al., 2002). GNAQ activates the IP3 calcium (Ca2+) signaling pathway and releases Ca2+ intracellularly which results in ERK1/2 phosphorylation causing CREB activation (Marklund et al., 1993; Sgouras et al., 1995; O’Donovan et al., 1999; Alvarez-Curto et al., 2016). Increased Ca2+ in GCG signaling is mediated by activated ADCY coupled to GLP-1R and GCGR. This activates PKA and phosphorylates Ca2+/calmodulin-dependent kinase II (CAMKII) in an IP3R- and Ca2+-dependent manner (Semizarov et al., 1998; Holz et al., 1999; Wang et al., 2000; Wu et al., 2002; Ding et al., 2004; Guan, Hou & Ricciardi, 2005; Glesne & Huberman, 2006; Diskar et al., 2010). Calmodulin (CALM) is a Ca2+ binding messenger activated upon binding of intracellular secondary Ca2+ and primarily transduces the Ca2+ signal in pancreatic cells (Chattopadhyaya et al., 1992; Tsang et al., 2006). Activation of CAMKII promotes FOXO1 nuclear translocation and plays a role in hepatic glucose production in response to fasting (Chattopadhyaya et al., 1992; Shaodong et al., 1999; Tsang et al., 2006). Hepatic expression of SMEK1/2 is also up-regulated during fasting, which is a key regulator of gluconeogenesis and elevates plasma glucose levels (Chowdhury et al., 2008). This causes dephosphorylation of CRTC2 which is responsible for transcriptional activation of gluconeogenic genes (O’Donovan et al., 1999; Iourgenko et al., 2003; Chowdhury et al., 2008).

Overview of SST signaling pathway. SST is a hormone that is involved in the inhibition of endocrine secretions such as INS, GCG, gastrin, and growth hormones as well as cell proliferation (Yamada et al., 1992; Pasquali et al., 2001; Luque & Kineman, 2018). SST binds to GPCRs called somatostatin receptors (SSTRs), of which there are five isoforms (SSTR1 to SSTR5) (Pasquali et al., 2001). Each receptor has distinct functions, and upon activation induce cascades of signaling pathways including protein tyrosine kinase activity (Yamada et al., 1992; Panetta et al., 1994). Upon tyrosine kinase stimulation, cytoplasmic protein-tyrosine phosphatase SHP1 is activated which triggers anti-proliferative and pro-apoptotic signals such as NF-kB, P53/Bax, and JNK (Sluss et al., 1994; Lisnock et al., 2000; Miao et al., 2000; Keilhack et al., 2001). Meanwhile, activation of SHP2 dephosphorylates the P13K/AKT and MEK pathways (Jones et al., 1991; Marklund et al., 1993; Keilhack et al., 2001; Zhang et al., 2006). This causes inhibition of cell proliferation via upregulation of P27, P21 cyclin kinase inhibitors, and the Zac1 tumor suppressor gene (Harper et al., 1993; Schneider, Montenarh & Wagner, 1998; Ishida et al., 2000). Pancreatic δ-cells secrete SST in response to elevated extracellular glucose concentrations (Hauge-Evans et al., 2009) and, within the islets, SST acts as a paracrine inhibitor of INS and GCG secretion (Hauge-Evans et al., 2009). SST is also a hypothalamic peptide known to inhibit somatic growth by inhibiting pituitary growth hormone (Hauge-Evans et al., 2009; Stengel, Rivier & Taché, 2013).

Introduction to corals

Corals are colonial marine invertebrates (cnidarians) that depend on a symbiotic relationship with dinoflagellate algae of the family Symbiodiniaceae (LaJeunesse et al., 2018). The algae harvest light and synthesize nutrients in exchange for shelter and nitrogen sources (Putnam et al., 2017). Coral reefs cover only 0.1% of the ocean floor but are home to the largest density of animals on earth, rivaling rain forest habitats in species diversity (LaJeunesse et al., 2018). The symbiosis, which was originally thought to be restricted to algae, is now known to extend to a much more complex community than anticipated with thousands of bacteria, bacteriophages, viruses, and fungi, in addition to endosymbiotic algae (Bourne et al., 2009). The entirety of the organism community in a coral is referred to as a holobiont, while the individual cnidarian host animals forming the colonies are called polyps.

The holobiont is characterized by balanced host-microbe molecular interactions. The complexity of these interactions in relation to stress and disease resistance, and recovery grow with every new study as questions arise regarding what molecules are responsible for symbiosis establishment and partner coexistence (Ainsworth & Gates, 2016; Kelly et al., 2021). These inter-partner exchanges are still poorly understood, and this is a particularly severe gap in our knowledge, since it is at the heart of the worldwide phenomenon of coral reef bleaching which refers to the breakdown of symbiosis (particularly the cnidarian host and endosymbiotic algae) due to thermal stress and high irradiance, including that brought about by global climate change. A recent study assessed 100 worldwide locations and found that the annual risk of coral bleaching has increased from an expected 8% of locations in the early 1980s to 31% in 2016 (Hughes et al., 2018; IPCC, 2022). Human impacts on coral reef ecosystems threaten fisheries and tourism, industries valued at hundreds of billions of dollars annually (Putnam et al., 2017). We are in urgent need of innovative solutions to increase corals’ resiliency to anthropogenic activities and facilitate their survival.

Climate change driven coral bleaching has now been recognized as the leading cause of the worldwide decline of coral reef cover and, overall, the biggest threat to reef-building coral survival (Hughes et al., 2017). Mass bleaching events have increased both in frequency and severity since the first recorded event in the 1980s (Oliver, Berkelmans & Eakin, 2018) and show no signs of reprieve as ocean warming gets compounded with traditional climate patterns such as the El Niño-Southern Oscillation (McPhaden, Zebiak & Glantz, 2006). Coral bleaching is the common term used to describe dysbiosis in the coral holobiont, specifically, the breakdown of symbiosis (xenophagy and/or expulsion) between the cnidarian host and the dinoflagellate endosymbionts (i.e., dinoflagellates provide most of the coral tissue pigmentation and as dysbiosis progresses, the tissue becomes transparent, thereby revealing the white calcium carbonate skeleton) (Suggett & Smith, 2020). While the full signaling cascade leading to dysbiosis is still poorly defined, we know it leads to damage to cell membranes, lipids, proteins and DNA via nitro-oxidative stress (i.e., the accumulation of free radicals, reactive oxygen species and reactive nitrogen species), a failing antioxidant machinery (e.g., catalase, ascorbate peroxidase, superoxide dismutase) and the organisms’ innate immune response (Weis, 2008; Lesser, 2011; Suggett & Smith, 2020).

Nitro-oxidative stress is common across aerobic systems and, in the case of the coral holobiont, has been associated with heat-damaged chloroplasts (Tolleter et al., 2013; Alderdice et al., 2022) and other damages to the photosynthetic mechanism through heat and light (Gleason & Wellington, 1993; Lesser & Farrell, 2004; Tolleter et al., 2013; Downs et al., 2013; Alderdice et al., 2022), the composition of the thylakoid membrane lipids (Tchernov et al., 2004), the potential for upregulation of ROS scavenging capacity and molecular chaperons during periods of thermal-, light-, or osmotic stress and hypoxia (Gardner et al., 2016; Levin et al., 2016; Ochsenkühn et al., 2017; Aguilar et al., 2019; Alderdice et al., 2021, 2022), seawater trace metal concentrations (Shick et al., 2011; Ferrier-Pagès, Sauzéat & Balter, 2018; Biscéré et al., 2018; Reich et al., 2023) and N:P ratios (Fabricius et al., 2013; Pogoreutz et al., 2017). Furthermore, bleaching has also been associated with the seawater carbonate saturation horizon and dissolved CO2 levels (i.e., ocean acidification) (Anthony et al., 2008; Crawley et al., 2010).

In the context of coral bleaching induced by heat stress, a study involving the tropical sea anemone Aiptasia pallida identified over 500 up-regulated genes, categorized into Cluster I linked to immunity and apoptosis and Cluster II related to protein folding, with potential regulators influenced by transcription factors NFκB and HSF1. A total of 337 genes in symbiotic anemones exhibited declining expression levels before visible bleaching, suggesting their involvement in algal symbiosis loss (Cleves et al., 2020). These findings hint at potential interactions of these genes with the INS signaling pathway, considering known roles of INS signaling in apoptosis and immune responses (Yuyama et al., 2018). Furthermore, experiments inducing ERK activity in corals via UV radiation and thermal stress (Courtial et al., 2017) and heat-shock experiments on Aiptasia (Sloan & Sawyer, 2016) contribute to our understanding of ERK and AKT phosphorylation and MAPK activities in these organisms, potentially implicating the INS signaling pathway in coral bleaching.

Glucose regulation in corals: an opportunity for understanding INS action in non-model organisms

The symbiotic algae provide as much as 90% of the energy corals consume by light harvesting and photosynthesis (Gierz, Forêt & Leggat, 2017). Thus, corals must be able to measure and regulate nutrient balance (Cunning et al., 2017). Given the crucial role of INS signaling for this task in other organisms, we here hypothesize that INS signaling may also exist in corals, although this hypothesis is purely theoretical and remains to be experimentally validated. Support for this hypothesis comes from transcriptomic studies (Yuyama et al., 2018). A comparison between the expression of INS signaling related genes in the presence and absence of the symbiotic algae strongly suggests that INS signaling is induced at the transcriptomic level in response to algal density in the tissue. A likely interpretation of this finding is that corals need to respond to the sugars produced by the algae through light harvesting and perhaps too much sugar could have detrimental effects on corals, similar to the diabetic response through aberrant INS signaling in humans. The symbiotic interaction between algae and coral involves algae entering the host, and the facilitation of energy and metabolite exchange. Algae utilize seawater substrates to synthesize a spectrum of organic compounds, effectively transferring vital nutrients, including amino acids, small peptides, sugars, carbohydrates, and lipids, to coral cells with glucose being a major metabolite transferred in this exchange as demonstrated by Burriesci and colleagues (Burriesci, Raab & Pringle, 2012). It is also possible that the mechanism for bleaching (loss of symbiotic algae from the coral holobiont) involves an imbalance in nutrient regulation and possible involvement of the INS signaling pathway. This raises an interesting speculation: could corals have diabetes, and could insulin resistance be related to the bleaching that is threatening coral species survival? While corals of course do not have Langerhans islets nor blood, the diabetes analogy at the molecular level may stimulate new ways of thinking about coral and human health (see below).

Indeed, there is evidence for INS signaling in corals at the molecular level. First, remote homology detection using HHblits have identified homologues for human IR and INSR in corals (Roger et al., 2022). HHblits is a so-called Hidden Markov Model (HMM)-based alignment approach developed by Remmert et al. (2011). Unlike traditional HMM profiles, in HHblits, both query and template are HMMs. The search for homologues is through an HMM-HMM alignment and the query HMM is generated by using amino acid distributions which makes this method extremely sensitive. It has been shown that, in many instances, HHblits successfully outperforms the identification and alignment of remote homologues, as compared to the traditional profile HMM approach, such as HMMER3 (Remmert et al., 2011). Given the 700 million years of evolution between corals and humans, this enhanced sensitivity of HHblits is instrumental to the comparison between corals and humans. We have already described the sequence alignments of the ligand-receptor pair, INS with INSR for human and for coral (Roger et al., 2022). In both cases, the alignments were identified with high confidence and cover a large fraction of the sequences: 1,164 out of 1,382 amino acids in the case of INSR and 101 out of 110 in the case of INS. The comparison of the sequences of human insulin (UniProt ID P01308) and Pocillopora damicornis (pdam) protein pdam_00006633, and the extracellular domain of the human INSR (UniProt ID P06213) and pdam_00013976 are shown in Tables 3A and 3B, respectively.

Table 3 Comparison of the sequences of human insulin and coral insulin.

Sequence comparison of human and Pocillopora damicornis insulin and insulin receptor sequences. A. Comparison of the sequences of human and coral INS. B. Matching residues in human INSR (visible in the 6pxv structure) with corresponding residues in coral INSR.

A.	
Amino acid sequence (human INS)	Amino acid sequence (coral INS)	
ALWMRLLPLLALLALWGPDPAAAFVNQHLCGSHLVEALY
LVCGERGFFYTPKTRREAEDLQVGQVELGGGPGAGSLQPLAL
EGSLQKRGIVEQCCTSICSLYQLENYC	LLWTIVPFLAIVLSLEAVTGSKLVKAYEVGSRRIDAHIC
GDHIKEVYTKVCIDESVGKRKRRSPLMEEKEALSFIHSE
SNRSLRKARSVRTVNIVEECCIEGCTIGELKEYC	
B.	
Domain	Amino acid sequence (human INSR) 6pxv	Amino acid sequence (coral INSR)	
L1	HLYPGEVCPGMDIRNNLTRLHELENCSVIEGHLQILLMF
KTRPEDFRDLSFPKLIMITDYLLLFRVYGLESLKDLFPN
LTVIRGSRLFFNYALVIFEMVHLKELGLYNLMNITRGSV
RIEKNNELCYLAT	VLKISNEKCDGCEKLENCTTLEGSIQVQMVRKASDAVMK
QLQFPKLTEITGHLLVSLMYGRRSLREIFPNLAVIRGRQ
VFLDYSLIIYQNDGLEEVNLPSLTTILRGGVRIEKNINL
CYVET	
CR	IDWSRILDSVEDNYIVLNKDDNEECGDICPGTNCPATVI
NGQFVERCWTHSHCQKVCPTICKSHGCTAEGLCCHSECL
GNCSQPDDPTK	IRWKSIMRNTKVDEYTLVLNSNNNDCYDRCFQQKCTPPA
GHGSLTNQYCWAPGAGSNADCQALCDMKCGDSGCVNGGL
MGKSTSCCDKQCLGGCTKTNSPHH	
L2	CVACRNFYLDGRCVETCPPPYYHFQDWRCVNFSFCQDLH
HKCKNSRRQGCHQYVIHNNKCIPECPSGYTMNSSNLLCT
PCLGPCPKVCHLLEGEKTIDSVTSAQELRGCTVINGSLI
INIRGGNNLAAELEANLGLIEEISGYLKIRRSYALVSLS
FFRKLRLIRGETLEIGNYSFYALDNQNLRQLWDWSKHNL
TITQGKLFFHYNPKLCLSEIHKMEE	CYACRNFRMPKGECVEKCGPGLYEIDEFKCIDNCPDGYL
KLGMKCAKVCPAGYKEGGNKSCLKCTTEKCPRGIGTQLE
ENLGQIEKVNGYIVIIESASLTSLNFFKNLREIRPRLIY
NFLSRPPAMETDLYNERYALAIRDNPKLEALWPFQQNLT
IIEGGIMVHLNPYLCPSQI	
FN3-1	VSGTKGRQERNDIALKTNGDQASCENELLKFSYIRTSFD
KILLRWEPYWPPDFRDLLGFMLFYKEAPYQNVTEFDGQD
ACGSNSWTVVDIDPPLRSNDPKSQNHPGWLMRGLKPWTQ
YAIFVKTLVTFSDERRTYGAKSDIIYVQTDAT	TPLINDILKWNRNDSNRVLDISDTTNGNAVACNVRKINV
TVEEITLPRGCNPVCVKVEWDDAIINDDYRNVLFYTLSY
REAPNRQITEYTDVDACSSDSGDIWTRIDHTVPPPEVNV
SRGLITKRRKIERTIKKLKPYQLYAFQVEAVVLKNDGAK
SDLVFVMTKESK	
FN3-2	NPSVPLDPISVSNSSSQIILKWKPPSDPNGNITHYLVFW
ERQAEDSELFELDYCLKGLKLPSR*QILKELEESSFRKTF
EDYLHNVVFVPR*EEHRPFEKVVNKES	PSQPVGLEANYLNSSALLVTWEPPLFPNGNITKYIVSYE
ISTYSAWKADLDWCSRQVFSNRL*EKMKPEKQSALFAKEF
QDILYKTLFTK*TKPNASLTVDGNVNKIP	
FN3-3	LVISGLRHFTGYRIELQACNQDTPEERCSVAAYVSARTM
PEAKADDIVGPVTHEIFENNVVHLMWQEPKEPNGLIVLY
EVSYRRYGDEELHLCVSRKHFALERGCRLRGLSPGNYSV
RIRATSLAGNGSWTEPTYFYVTDYLD	LTNLRHFSDYTITVCACTKVGCATGSSCATTKGMTNKNG
SRQIIKIFLCIISATVSIIMGFPKGPKWSGAQSDSQPEF
KCVSGKELKYQEKVEDGNYSAQVRAITSSGNGSWSNTVS
FSYFIESQSTVPPIGE	
Note:

* Indicates missing sequence in the structure.

The finding of a human INS homologue in Pocillopora damicornis has prompted us to test the effect of human INS on corals experimentally (Roger et al., 2022). An average 20% reduction in viability at 100 µg/mL INS concentration was observed in line with its proteotoxicity in other systems (Rege et al., 2020). Due to the importance of INS administration in diabetes, its folding and stability has been studied extensively (Weiss & Lawrence, 2018; Liu et al., 2018). High concentrations of salts are known to promote INS aggregation and misfolding (Grudzielanek et al., 2007; Chatani et al., 2014), and the use of seawater in our experiments may induce similar effects, which may be the cause for the observed cytotoxicity.

As shown in Fig. 2, not only INS and INSR, but a total of 75 proteins (excluding isoform counting) are involved across the three INS related pathways in humans. Application of the non-model organism pipeline described above (Kumar et al., 2023) reveals that the majority of downstream signaling proteins, namely 67 of the 75 human proteins, are likely conserved in Pocillopora damicornis. In Fig. 2, all human proteins shown in black have a predicted Pocillopora damicornis homologue, while those shown in red do not. Crosstalk between the SST, CGC and INS pathways is mediated by several proteins that are common to two or even all three pathways. We were not able to identify suitable Pocillopora damicornis homologues for eight proteins: GLP-1, GLP-1R, CGC, CGCR, SST, SSTR, BAD, and CALM, as judged by their poor e-values as well as low percent alignments of amino acids. It is important to realize that GLP-1R, CGCR and SSTR are all GPCR’s and thus it is difficult to differentiate GPCR variation within organisms as compared to across organisms. This complication has been discussed in detail, and it was proposed that the GPCR repertoire of Pocillopora damicornis is 151 as compared to 825 in human (Kumar et al., 2023). The results of the remote homology search can be accessed through supplementary file S4 of that article, where all three human GPCR sequences (CGCR, SSTR and GLP-1R) are never ranked first for any of the coral GCPR candidates. The most closely related GPCR is pdam_00008152-RA, which is more similar to the GLP-2 receptor than those three. Thus, it may be possible that GLP-2 is a more ancient modulation of the INS pathway than the GLP-1, CGC and SST pathways (Amato, Baldassano & Mulè, 2016). This finding suggests that the lack of interest in GLP-2 in previous human studies reviewed above is perhaps unjustified. It is important to note that these new evidences are computational only and await future experimental validation. While SST, GLP-1 and GCG are the ligands initiating their respective signaling pathways through their respective GPCR’s, BAD is located at the effector end of the main INS pathway, indicating that this pathway is mostly functional. Similarly, CALM functions only to stimulate PYGL in the GCG signaling cascade, and TSC1/2 in INS signaling. Importantly, there is a clear homologue of both INS and the INSR present.

Outlook: coral pharmacology

In this review, we have pointed out numerous possible analogies between human and coral INS biology. The enormous pharmacological importance of treating INS resistance in diabetes makes it tempting to speculate that we can translate what we know about human INS pharmacology to corals, coining a new field of “coral pharmacology” which opens the door to thinking about drug discovery and treatments for corals. While at present we do not know if and how we can deliver medicines to corals in the vast ocean in practical terms, ideas include coating surfaces on which coral larvae settle, or feeding corals or dispersing such compounds into the ocean in proximity to coral reefs. This is not unthinkable given that there are many known examples of small molecules secreted into the ocean used in communication between different inhabitants of a reef, e.g., for attracting fish to anemones (Kamio & Derby, 2017; Saha et al., 2019; Kamio, Yambe & Fusetani, 2022; Morgan et al., 2022) or to mediate biological interactions with surfaces during settling of coral larvae (Petersen et al., 2023). Nanocarriers could also assist with this purpose (Roger et al., 2023). Given the fluid nature of the ocean environment, such small molecules can be dispersed easily and thus the environmental impact of treatment of corals with small molecule “coral drugs” will need to be carefully addressed. Nonetheless, the idea of “coral pharmacology” may open new avenues to think about how to tackle the coral bleaching crisis. How might coral drug discovery look like? The high quality of sequence alignments of INS and INSR with respective coral homologues shown in Table 3 provides an opportunity to exploit the large amount of INSR structural data that has become available recently (McKern et al., 2006; Menting et al., 2013; Gutmann et al., 2018; Weis et al., 2018), especially due to the advances in cryoelectron microscopy (Uchikawa et al., 2019). Shown in Fig. 3 are homology models for the various domains in human used to predict coral INSR using the sequences shown in Table 3B. These models open the door to the first step of future exploration of potential drug targets in coral, exemplified here by the INSR as a drug target, extrapolated from its role as a human drug target (Kumar, Vizgaudis & Klein-Seetharaman, 2021). The concept of “coral pharmacology” aims to develop pharmacological approaches towards potentially treating corals who have been harmed by human activities. Using membrane receptors as a proof of concept, we developed a pipeline for establishing the functional similarities between human and coral membrane receptor signaling systems (Kumar et al., 2023). This pipeline extends to the INS-INSR pair and its related signaling pathway (Fig. 2). Given the role of INS signaling for regulation of nutrient concentration in humans, we surmise that the coral homologues will likely carry out a similar function in corals. This suggests that early metazoans such as corals use the INS system despite their simple organization. This may have major implications for coral bleaching and the communication across cnidarian host and symbiotic algae. Transcriptomic analysis has revealed that INS signaling is clearly affected by the establishment of symbiosis between cnidarian host animals and algal symbionts (Yuyama et al., 2018). Given that one of the major benefits of symbiosis is the delivery of sugars obtained through photosynthesis of the algae to the host, we can expect that the role of INS signaling is analogous in corals to that in humans, despite their evolutionary distance. It is tempting to speculate that under high light conditions, when the algae synthesize excess sugars, that the cnidarian host may experience INS resistance, a hypothesis that remains to be validated experimentally. By inference, pharmacological treatment of INS resistance may allow coral rescue. We have already shown that human INS does have an effect on corals (see above), and in fact is cytotoxic (Roger et al., 2022). The presence of receptors such as IR (described here) and GPCR (described in Kumar et al., 2023) suggests that many other functions of corals could potentially be targeted by pharmacological means to help prevent their extinction predicted under current climate trajectories.

Figure 3 Structure prediction of coral insulin receptor.

3D reconstruction of coral INSR. (A) Side view. (B) Top view. (C) Bottom view. (D) Individual domains of coral INSR. Created with the PyMOL Molecular Graphics System, Version 2.5 Schrödinger, LLC.

Additional Information and Declarations

Competing Interests

Author Contributions

Data Availability

The authors declare that they have no competing interests.

Meghana Hosahalli Shivananda Murthy conceived and designed the experiments, performed the experiments, analyzed the data, prepared figures and/or tables, authored or reviewed drafts of the article, and approved the final draft.

Paniz Jasbi conceived and designed the experiments, performed the experiments, analyzed the data, prepared figures and/or tables, authored or reviewed drafts of the article, and approved the final draft.

Whitney Lowe conceived and designed the experiments, performed the experiments, prepared figures and/or tables, and approved the final draft.

Lokender Kumar analyzed the data, prepared figures and/or tables, and approved the final draft.

Monsurat Olaosebikan analyzed the data, prepared figures and/or tables, and approved the final draft.

Liza Roger analyzed the data, authored or reviewed drafts of the article, and approved the final draft.

Jinkyu Yang analyzed the data, authored or reviewed drafts of the article, and approved the final draft.

Nastassja Lewinski analyzed the data, authored or reviewed drafts of the article, and approved the final draft.

Noah Daniels analyzed the data, authored or reviewed drafts of the article, and approved the final draft.

Lenore Cowen analyzed the data, authored or reviewed drafts of the article, and approved the final draft.

Judith Klein-Seetharaman conceived and designed the experiments, performed the experiments, analyzed the data, prepared figures and/or tables, authored or reviewed drafts of the article, and approved the final draft.

The following information was supplied regarding data availability:

This is a literature review.

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
