# Peer review of "Insulin signaling and pharmacology in humans and in corals"

_PeerJ, doi:10.7717/peerj.16804_

## Round 0.1 · original submission · Minor Revisions

Dear Dr. Klein-Seetharaman,

Thank you for submitting your manuscript to PeerJ.
I am pleased to note that the reviewers have made largely positive comments regarding the manuscript's content and structure.Nevertheless, there are a few minor points that require attention.

Thank You
Sushanta

Reviewer 1 ·

Basic reporting

This is a well written paper with relevant information.

Experimental design

Study design iss very appropriate and also meet the Aims and scope of the journal.

Validity of the findings

Conclusion are well stated and very concise.

Additional comments

This is well designed and well written paper you can submit this with no correction.

·

Basic reporting

1. Use of professional and unambiguous English language throughout the review enables to gain a clear understanding of the highlighted and discussed issue.

2. Very informative and crisp introduction that enables the reader to get a quick grip on the subject discussed later.

3. The subject discussed in the review has been very carefully chosen. The problem of coral bleaching has been quite a menace for the entire scientific community for the past few decades. Apart from the fact that coral bleaching is caused by anthropogenic activities, very little is known about the molecular phenomena underlying this problem.

Experimental design

Survey methodology is very elaborate and informative.

Validity of the findings

A very different but logical angle of looking at the whole scenario. This review highlights a very important point: structural homology between the otherwise evolutionarily distant human INS/INSR and coral INS/INSR. Also, another important finding highlighted here is that corals harbor 67 out of the total 75 proteins involved in the three insulin metabolism pathways in humans. This is a very important finding and opens up avenues for further research in the field of coral insulin metabolism and coral pharmacology. Also, this finding shows an entirely new angle of looking at the current problem of coral bleaching.

Additional comments

The review would be more informative if the writers incorporated the following points:

1. Please add a separate paragraph about factors responsible for coral bleaching in detail:

1. High radiation
2. Thermal stress
3. Excess buildup of Nitrogen
4. Deoxygenation ( REF: Alderdice R, Perna G, Cárdenas A, Hume BC, Wolf M, Kühl M, Pernice M, Suggett DJ, Voolstra CR. Deoxygenation lowers the thermal threshold of coral bleaching. Scientific Reports. 2022 Oct 31;12(1):18273.)

Concept of Coral diabetes or coral insulin tolerance as one of the reasons for “coral bleaching” appears interesting enough to be discussed. But, on the contrary there are no studies to support the above hypothesis that aberrant insulin signaling is the reason behind coral bleaching.
However, in contrast to the above, studies have shown aberrant expression of many stress responsive genes in response to heat stress exposure. These genes were found to be associated with a number of vital functions like innate immune response and protein homeostasis (REF: Cleves PA, Krediet CJ, Lehnert EM, Onishi M, Pringle JR. Insights into coral bleaching under heat stress from analysis of gene expression in a sea anemone model system. Proceedings of the National Academy of Sciences. 2020 Nov 17;117(46):28906-17.)

2. The review quotes a statement from a published article, "Toxicity of human insulin to corals (at the concentration 33.9 µg/mL) as pointed out in the study by Roger et al 2022 (REF: Roger LM, Adarkwa Darko Y, Bernas T, White F, Olaosebikan M, Cowen L, Klein-Seetharaman J, Lewinski NA. Evaluation of fluorescence-based viability stains in cells dissociated from scleractinian coral Pocillopora damicornis. Scientific reports. 2022 Sep 12;12(1):15297) and existence of a structurally similar molecular that sustains the coral homeostasis" is a bit confusing. Please discuss it in detail for a clear understanding.

·

Basic reporting

No comment

Experimental design

No comment

Validity of the findings

This article is well conceptualized and written accordingly, yet the article needs to include more studies explaining the similarities between coral glucose metabolism and human INS signaling. Sequence conservation between organisms is well reported, i.e., no random human gene will show sequence similarity less than 30% with other organisms. Therefore, it is a bit stark to consider a systems orthology based only on sequence similarity. It would be better if more articles showing the connection between human INS signaling and coral glucose metabolism were included in the study.

Major Comments:

1. The human INS signaling pathway is well explained but coral metabolism remains poorly understood.

2. Add more studies explaining the homology between the human and coral INS signaling pathways. A table including genes associated with INS signaling along with their sequence similarity in both organisms can be included in this manuscript.

3. Coral pharmacology is a good concept that should be pursued, yet it has no relevance in practical terms. Can you please explain this a little.

Minor comments:

1. Please check the references thoroughly.

For example: Pedersen et al. (line no. 789) and Sadzikowska et al., (line no. 825) are not in the text but are included in the references section.

References in the table (e.g. references 155, 160) are not included in the references section.

---

## Round 0.2 · accepted · Accept

Dear Dr. Klein-Seetharaman

Thank you for submitting your work to PeerJ. The reviewers have expressed satisfaction with the responses addressing the specific points raised in the manuscript. The current version of the manuscript is deemed suitable for publication.

·

Basic reporting

Clear and unambiguous english used throughout the article.

Sufficient literature references have been provided

Professional article structure

Introduction adequately introduces the subject, which has later been justified in the later part of the review

Experimental design

Study design is adequate and within the journal guidelines.

Survey methodology is consistent and unbiased

Review is well-structured and organized into clear paragraphs

Validity of the findings

The findings are valid, well stated and meet out the goals stated in the introduction.

·

Basic reporting

The manuscript is well organized so is self-explanatory and matches the scope of the journal.

Experimental design

No comments

Validity of the findings

More dedicated work towards coral pharmacology might open the way for homology based interdisciplinary studies in the future.

Additional comments

The concept of coral pharmacology is new yet relevant and can prove to be an important step in saving coral diversity that is being destroyed.